# SR-PFN: Yet Another Sequential Recommendation Paradigm

## Abstract

Sequential recommendation is a popular task in many real-world businesses. On the one hand, conventional sequential recommenders learn collaborative signals and temporal patterns solely from training interactions and do not generalize well to new datasets. On the other hand, to better leverage textual metadata and user reviews, LLM-based recommenders have recently been proposed; however, they often incur high inference costs and may inherit limitations of language models, including limited multilingual generalization, social bias, and a tendency to memorize data rather than to infer. To this end, we present SR-PFN, a sequential recommender that performs single-pass next-item prediction via in-context inference after being pretrained on synthetic data — our method is the first attempt for sequential recommendation under the regime of Prior-data Fitted Networks (PFNs). Our approach introduces a synthetic prior model tailored toward sequential recommendation. After being pre-trained on synthetic data sampled from the prior model, which reflects realistic sequential dynamics, SR-PFN learns to approximate the posterior predictive distribution (PPD) for next-item prediction at test time, enabling parameter update-free, single-pass inference. Across sequential recommendation benchmarks, SR-PFN outperforms seven competitive baselines, while offering substantially lower inference costs compared to those of LLM-based models.

## 1 Introduction

Sequential recommendation (Wang et al., 2019; Fang et al., 2020) aims to predict the next item that a target user will interact with based on their interaction history. Existing sequential recommender systems learn embedding representations by extracting collaborative and sequential patterns directly from observed user-item interaction histories to capture user preferences (Hidasi et al., 2015; Kang & McAuley, 2018; Sun et al., 2019). More recently, large language model (LLM)-based approaches (Geng et al., 2022; Bao et al., 2023) have emerged, where map items into a natural language embedding space and model sequential dynamics using pretrained textual representations.

While user-item interaction histories provide the empirical basis for sequential recommendation, relying only on such histories introduces practical challenges. These challenges restrict models to learning rather simple patterns present in the training data, hindering their ability to generalize to other datasets (Zhu et al., 2021; Zang et al., 2022). They also require dataset-specific retraining or extensive re-tuning to transfer across domains, which raises operational costs. To better leverage textual metadata and user reviews, recent attempts to utilize LLMs have shown promising accuracy (Kong et al., 2024; Kim et al., 2025), but often incur prohibitive inference costs and latency, limiting their practicality in real-world deployment scenarios. They may also inherit limitations of LLMs, which primarily focus on English tasks (Zhang et al., 2020), and exhibit social bias (Gallegos et al., 2024), as well as a tendency to memorize training data (Di Palma et al., 2025).

In this work, we propose **SR-PFN** — a new paradigm for sequential recommendation built on Prior-Data Fitted Networks (PFNs; Müller et al., 2022). SR-PFN is pretrained once under synthetic data sampled from a prior data distribution, then infers the posterior predictive distribution (PPD) of each query from in-context examples without any parameter updates. Figure 1 shows how SR-PFN differs from conventional sequential recommenders. During the pretraining stage of SR-PFN, the model learns to infer based on a diverse spectrum of interaction patterns from in-context examples,

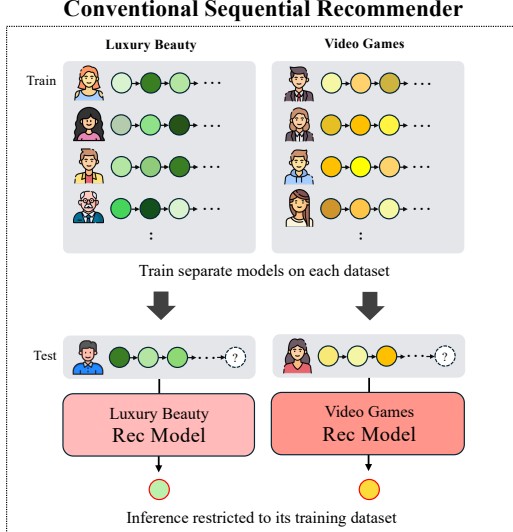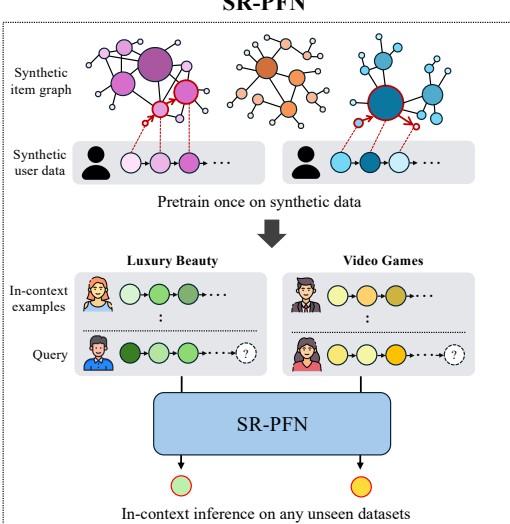

Figure 1: Left: Conventional sequential recommenders are trained separately on each real dataset (e.g., Luxury Beauty, Video Games) and can only be applied within the same domain, so their inference remains confined to the training set. Right: SR-PFN is pretrained once on diverse synthetic data that capture a wide range of interaction patterns, and then performs in-context inference on unseen datasets using only a few example sequences from the target domain — without any retraining or fine-tuning. In all graphs, node size reflects item popularity.

thereby enabling it to make contextually relevant predictions on new tasks by processing examples from real-world datasets in-context, without requiring data-specific training. While prior PFN research has demonstrated strong performance in other areas such as tabular classification (TabPFN; Hollmann et al., 2023) and time-series forecasting (ForecastPFN; Dooley et al., 2023), to the best of our knowledge, SR-PFN represents the first attempt to extend PFNs to the domain of sequential recommendation.

A core challenge of this paradigm is to design a prior that reflects the statistical properties observed in real-world sequential recommendation data. Our prior model (see Section 3.2) works in the following two steps: i) grounded in the observation that user-item interaction histories exhibit an item-item co-occurrence structure with heavy-tailed degrees and a hierarchical community organization (Yang & Leskovec, 2012; Abdollahpouri et al., 2019), we adopt a hierarchical degree-corrected stochastic block model (hDCSBM; Karrer & Newman, 2011; Peixoto, 2014) prior as a parametric prior of item graphs; ii) on top of the hDCSBM-generated item-item graph, we generate interaction sequences using Personalized PageRank (PPR; Haveliwala, 2002), which serve as in-context examples and queries for pretraining SR-PFN for the next-item prediction task.

The next challenge is the model architecture. For SR-PFN to capture interaction patterns, we first propose a gated fusion encoder that integrates low-rank embeddings derived from the user–item interaction matrix and the item–item transition matrix, constructed from user interactions sampled from the same prior (see Section 4). On top of these representations, we design a prompt tailored to in-context learning (Brown et al., 2020), consisting of $k$ example blocks and one query block. A block-aware attention mask is applied to the prompt, allowing for a query-to-example flow while preventing cross-example and answer-to-query leakage, thereby preserving temporal causality and block-wise independence.

Pretrained once under a synthetic prior, SR-PFN shows strong performance across standard sequential recommendation benchmarks. Even when existing recommenders are trained end-to-end on the observed interactions to directly learn embeddings and relations, SR-PFN, without any parameter updates, outperforms five ID-based and two LLM-based models while achieving 6x lower inference cost than the LLM-based approaches (see Section 5). By being trained under a broad synthetic prior that captures diverse interaction patterns, SR-PFN also demonstrates particular strength in cold-start scenarios where user histories or item interactions are sparse.

**Contributions**   We make the following contributions:

- We introduce SR-PFN, a PFN-based paradigm for sequential recommendation that performs in-context learning in a single forward pass, without any parameter updates.
- We design a controllable prior over item graphs via an hDCSBM coupled with PPR-based sequence generation, which captures salient statistics of sequence-derived co-occurrence graphs.
- We propose a gated fusion encoder that fuses low-rank embeddings from the user–item interaction and item–item transition matrices, together with a prompt and block-aware attention mask that learn sequential patterns from in-context examples and then transfers them to the query.
- Our SR-PFN demonstrates strong accuracy against 5 ID-based and 2 LLM-based baselines, with substantially lower inference cost than the LLM-based approaches.

## 2   RELATED WORK

**Sequential recommendation**   Traditional sequential recommenders identify users and items with unique IDs and learn embeddings from interaction sequences (Wang et al., 2019; Fang et al., 2020). Early statistical approaches include matrix factorization (MF) methods (Koren et al., 2009), which decompose the user–item interaction matrix to model collaborative signals, and Markov chain (MC) methods, which treat user histories as ordered sequences (Rendle et al., 2009; 2010). The rise of deep learning brought models such as GRU4Rec (Hidasi et al., 2015) and Caser (Tang & Wang, 2018), which leverage RNN and CNN architectures to model complex and nonlinear sequence patterns (Guo et al., 2017; Yuan et al., 2019). The introduction of attention mechanisms further led to models like SASRec (Kang & McAuley, 2018), which use self-attention to focus on the most relevant items in long user histories (Sun et al., 2019; Xie et al., 2022). However, these ID-based approaches often lack semantic understanding, resulting in limited suboptimal personalization (Yuan et al., 2023).

Recent research has explored leveraging LLMs' strong generalization and semantic understanding for recommendation by reformulating tasks as text prompts (Geng et al., 2022). Early approaches such as TALLRec (Bao et al., 2023) focused on fine-tuning with recommendation data to better align LLMs with recommendation objectives (Zhang et al., 2025a). Later methods move beyond fine-tuning by using hybrid prompting, which injects collaborative filtering embeddings into the LLM input space, allowing the model to exploit CF signals when generating recommendations (Liao et al., 2024; Kong et al., 2024; Zhang et al., 2025b; Kim et al., 2025). In-context learning (Brown et al., 2020) adapts models to new tasks from a few examples without parameter updates.

**Prior-data fitted networks**   Müller et al. (2022) showed that Prior-data Fitted Networks (PFNs) approximate Bayesian posterior predictive inference from in-context examples, and subsequent theory established why they succeed through bias–variance mechanisms (Nagler, 2023). Since then, PFNs have been extended beyond tabular and time-series tasks (Hollmann et al., 2023; Dooley et al., 2023) to domains such as biology (Ubbens et al., 2025; Scheuer et al., 2025), causal inference, and anomaly detection (Shen et al., 2025; Ma et al., 2025), often by designing synthetic priors that capture domain-specific structures. Recent work further addresses PFNs' in-context limitations through improved context selection and ensemble methods, advancing their scalability and generalization (Feuer et al., 2025; Wang et al., 2025; Müller et al., 2025).

## 3   PRIOR FOR SEQUENTIAL RECOMMENDATION

### 3.1   BACKGROUND ON PRIOR-DATA FITTED NETWORKS

Let $\Phi$ denote a hypothesis class of data-generating mechanisms. Each hypothesis $\phi \in \Phi$ defines a distribution over user–item interactions and thereby generates both (i) a dataset of in-context examples $D = \{(x_i, y_i)\}_{i=1}^n$, where $x_i$ is a user history sequence and $y_i$ its ground-truth next item, and (ii) additional query pairs $(x_q, y_q)$ drawn from the same mechanism. For evaluation, each query user history sequence $x_q$ is accompanied by a candidate set $C_q$, constructed using a random negative sampling policy $\nu(C_q \mid D, x_q, y_q)$. Let $U^-(D, x_q)$ be the set of items not previously interacted with by $x_q$ among the dataset $D$. We assume (i) $C_q = \{y_q\} \cup S_q$ with $S_q \subseteq U^-(D, x_q) \setminus \{y_q\}$ and

$|C_q| = m$ fixed; (ii) conditional on $(D, x_q)$, $S_q$ is sampled uniformly without replacement from $U^-(D, x_q) \setminus \{y_q\}$.

$$\nu(C \mid D, x_q, y_q) = \frac{1}{\binom{|U^-(D,x_q)|-1}{m-1}}.$$

Throughout, $p(\cdot \mid D, x_q)$ denotes the Bayesian posterior predictive induced by a prior $p(\phi)$ over mechanisms $\phi \in \Phi$:

$$p(y_q \mid D, x_q) = \int p(y_q \mid x_q, \phi)\, p(\phi \mid D)\, d\phi, \qquad p(\phi \mid D) \propto p(D \mid \phi)\, p(\phi).$$

For a query user history $x_q$ with candidate set $C_q$, the Bayesian posterior predictive distribution conditioned on $C_q$ is

$$p(y_q \mid D, x_q, C_q) = \frac{p(y_q \mid D, x_q)\, \nu(C_q \mid D, x_q, y_q)}{\sum_{c \in C_q} p(c \mid D, x_q)\, \nu(C_q \mid D, x_q, c)}. \tag{1}$$

If negatives are sampled uniformly without replacement from the items not seen by $x_q$, then $\nu(C_q \mid D, x_q, y)$ is independent of which $y \in C_q$ is the ground truth. In this case, the $\nu$-factor cancels and the conditional reduces to a simple renormalization:

$$p(y_q \mid D, x_q, C_q) = \frac{p(y_q \mid D, x_q)}{\sum_{c \in C_q} p(c \mid D, x_q)}, \quad \text{where } y_q \in C_q. \tag{2}$$

Following the *synthetic prior-fitting* introduced in previous works (Müller et al., 2022; Adriaensen et al., 2023), training SR-PFN with cross-entropy on synthetic tasks yields a predictor $q_\theta$ that matches this candidate-restricted conditional. For each prompt $(D, x_q, C_q)$,

$$\mathbb{E}_{y_q \sim p(\cdot \mid D, x_q, C_q)}[-\log q_\theta(y_q \mid D, x_q, C_q)] = H(p) + \mathrm{KL}\big(p \parallel q_\theta\big), \tag{3}$$

so the unique minimizer is $q_\theta^\star = p(\cdot \mid D, x_q, C_q)$. When negatives are sampled uniformly without replacement from the unseen pool, that is, when the candidate set distribution follows the policy $\nu(C_q \mid D, x_q, y)$, the expression coincides with equation 2. For complete proofs and discussion, see Appendix B.

## 3.2 Generating Synthetic Data for Sequential Recommendation

The synthetic data generation mechanism $\phi \in \Phi$ specifies (i) how to construct an item graph, (ii) how to synthesize user sequences on that graph, and (iii) how to compute representation embeddings from those sequences.

**Common properties of real-world sequential interaction data**   Before discussing our framework for synthetic data generation, we first examine the common properties of real-world sequential interaction data.

- **Heavy-tailed item popularity** A small set of head items accounts for a disproportionately large share of interactions, while the majority of items lie in the long tail. This skew can be quantified by the *degree exponent*, estimated from the complementary cumulative distribution of item degrees, which captures tail heaviness (Clauset et al., 2009; Yin et al., 2012).
- **Head dominance** Complementary to the exponent, the *head fraction* measures the proportion of interactions accounted for by the top $q$ fraction of items (e.g., $q$=10% of the catalog). This statistic directly reflects the imbalance between head and tail usage (Abdollahpouri et al., 2017; Klimashevskaia et al., 2024).

Both quantities are dataset-dependent, and our synthetic prior exposes them as explicit parameters, enabling faithful reproduction of the skew observed in real domains.

**Hierarchical degree-corrected stochastic block model**   To model these properties, we adopt a hierarchical degree-corrected stochastic block model (hDCSBM; Karrer & Newman, 2011; Peixoto, 2014) as the backbone of our synthetic prior. The degree-correction mechanism enables us to impose long-tailed popularity directly. Each item is assigned a propensity drawn from a truncated

power law, resulting in a controllable degree exponent and head fraction. In parallel, the hierarchical structure organizes items at two levels of granularity: macro-communities (coarse groups) and micro-communities (finer subgroups). Each community is defined as a set of items that co-occur more frequently with one another than with the rest of the catalog. By organizing items into such communities with hierarchy, hDCSBM can mimic the hierarchical grouping observed in real datasets (e.g., product hierarchies or genres) while retaining explicit control over popularity skew. See Appendix C.1 for a detailed explanation of how we modeled community structures.

**Random walk with restarts (Personalized PageRank)**   On top of the weighted adjacency matrix generated by the hDCSBM, we row-normalize it to obtain a Markov kernel $K$, which specifies a random walk on the item graph. On $K$, we generate a synthetic interaction sequence for each user $u$ using Personalized PageRank (PPR; Haveliwala, 2002) — a user-conditioned random walk with restart that follows outgoing edges with probability $\alpha$ and teleports to a personalization distribution $\pi_u$ with probability $1 - \alpha$, where $\pi_u$ is a probability vector specifying the restart locations. We then draw a sequence length $\ell_u$ from a truncated power law to reflect heavy-tailed user activity. The user-specific PPR vector $p_u$ is the unique fixed point $p_u = \alpha K^\top p_u + (1 - \alpha) \pi_u$ interpreted as the stationary distribution of a restart random walk.

**Matching skew statistics in practice**   To assess how well the synthetic prior matches real-world data, Figure 2 compares two statistics between real datasets (colored markers) and our synthetic prior (gray boxplots): the *degree exponent* and the *head fraction*. For each of the five datasets from the Amazon 2018 corpus[1], we compute these statistics directly from their user sequences. For the synthetic side, we generate 100 independent catalogs by sampling an hDCSBM and then producing user sequences via the PPR. The gray boxplots summarize the resulting values across runs (the red line indicates the median). The synthetic distribution covers the empirical markers from real datasets, indicating that the hDCSBM-based mechanism

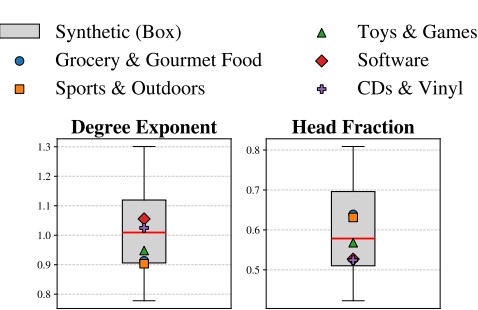

Figure 2: Matching skew statistics. Left: degree exponent estimated from the CCDF of item degrees; Right: head fraction (share of interactions explained by the top 10% most popular items).

can tune both statistics in a controlled manner, matching dataset-specific skew without overfitting to any single catalog. Detailed steps and hyperparameters for generating synthetic data for sequential recommendation are described in Appendix C.

**Low-rank representations of interactions and transitions**   Having specified the synthetic generative mechanism, we now derive low-rank representations that serve as inputs to SR-PFN. For each synthetic sequence $x_u = (i_{u,1}, \ldots, i_{u,\ell_u})$, we hold out the last item $i_{u,\ell_u}$ and use the history prefix $H_u = x_u[: -1]$ to construct two matrices. First, the user–item interaction matrix is defined as $X_{u,i} = \mathbb{I}\{ i \in H_u \}$, where $\mathbb{I}$ is the indicator function. Second, the item–item transition matrix $R$ is built from row-normalized bigram counts: for each pair $(i, j)$, we count how often $i \rightarrow j$ appears in $H_u$ across all users to form $B_{ij}$, then normalize rows to obtain $R = D_R^{-1}B$ with $D_R = \mathrm{diag}(B\mathbf{1})$. Because $R$ is row-stochastic, each entry $R_{ij}$ can be interpreted as the conditional probability $P(j \mid i)$, naturally aligning the representation with next-item prediction and improving numerical stability across tasks. Finally, we compute truncated SVDs of $X$ and $R$ and use the resulting low-rank user/item embeddings $(u, i)$ from $X$ and row/column embeddings $(r, c)$ from $R$ as model inputs.

## 4   SR-PFN: PFN FOR SEQUENTIAL RECOMMENDATION

SR-PFN operationalizes PFN-style inference for sequences via three parts: (i) a prompt that organizes in-context examples and a query under candidate lists, (ii) a block-aware attention mask that regulates information flow, and (iii) a lightweight encoder that fuses low-rank embeddings.

---

[1] https://cseweb.ucsd.edu/~jmcauley/datasets/amazon_v2/

| | | | | | | | | | | |
|---|---|---|---|---|---|---|---|---|---|---|
| Example 1, Block ID=1 | EXAMPLE_START | INTERACTION | ... | CONTEXT_END | CANDIDATE_START | CANDIDATE | ... | ANSWER | POS_ITEM | EXAMPLE_END |
| Example 2, Block ID=2 | EXAMPLE_START | INTERACTION | ... | CONTEXT_END | CANDIDATE_START | CANDIDATE | | ANSWER | POS_ITEM | ... EXAMPLE_END |
| ⋮ | | | | | ⋮ | | | | | |
| Example k, Block ID=k | EXAMPLE_START | INTERACTION | ... | CONTEXT_END | CANDIDATE_START | ANSWER | | POS_ITEM | ... CANDIDATE | EXAMPLE_END |
| | | | | | | | | | | |
| Query, Block ID=k+1 | QUERY_START | INTERACTION | ... | CONTEXT_END | CANDIDATE_START | CANDIDATE | ... | CANDIDATE | POS_ITEM | QUERY_END |

Figure 3: Prompt visualization with $k$ in-context example blocks followed by one query block.

## 4.1 PROMPT CONSTRUCTION

We serialize the demonstrations and the query into block-structured text so that the model can read the examples and answer a candidate-restricted query. Figure 3 shows serialization of $k$ in-context example blocks followed by a single query block. A block spans from EXAMPLE_START or QUERY_START token to EXAMPLE_END or QUERY_END token. Within each block, the user's history is a sequence of INTERACTION tokens; CONTEXT_END then marks the boundary to the candidate list, which begins at CANDIDATE_START. The single positive in each candidate set is the penultimate item of the underlying sequence, for example, and the final item for the query. Within example blocks only, we insert ANSWER immediately before the positive item, which is marked as POS_ITEM in Figure 3. The remaining $C-1$ candidates are sampled as uniform negatives without replacement. See Appendix D.1 for the full token summary and roles. In this work, we select $k \in \{0, 1, 2, 4, 8\}$ in-context examples. This choice is supported by prior findings showing that using only a subset of context examples most similar to the query can yield comparable or even better performance (Thomas et al., 2024; Ye et al., 2025). A detailed algorithm is provided in Appendix D.2, and the ablation study on the number of in-context examples is given in Appendix H.2.

## 4.2 BLOCK-AWARE ATTENTION MASK

To align attention with the prompt semantics while preventing leakage of query information to examples, we introduce a block-aware attention mask. Each token is assigned a block ID, which is incremented at every EXAMPLE_START or QUERY_START. Figure 3 illustrates this layout. The mask operates at two levels: inter-block and intra-block. For the inter-block policy, tokens in example blocks may attend only to their own and earlier blocks (no look-ahead across examples), whereas tokens in the query block may attend to all blocks. For the intra-block policy, we split each block into a history region and a candidate region at the CONTEXT_END token. History tokens use a left-to-right causal mask and cannot attend to candidates. CANDIDATE tokens have full attention within the same block (they may attend to the block's history and to other candidates) but never across blocks. The final attention mask is the intersection of the inter- and intra-block masks. Scoring and loss are computed only for the query candidates. See Appendix D.3 for visualization and description of the attention mask.

## 4.3 ENCODER

To integrate low-rank embeddings, we introduce an encoder that jointly fuses user, item embeddings $(u, i)$ derived from the interaction matrix and row, column embeddings $(r, c)$ obtained from the transition matrix. Figure 4 shows the overall architecture of the encoder, where boxes with black borders denote learnable components. Each embedding component is first rescaled by a learnable scalar weight, yielding $u', i', r', c'$, which allows amplification or attenuation of signals from the original embeddings. Each gate layer takes as input the concatenation of the two rescaled embeddings, e.g., $[u'\|i']$ or $[r'\|c']$, and processes them through a small network to generate feature-wise weights. Then these weights are applied element-wise ($\odot$) to the corresponding multiplicative views $u' \odot i'$ and $r' \odot c'$, producing

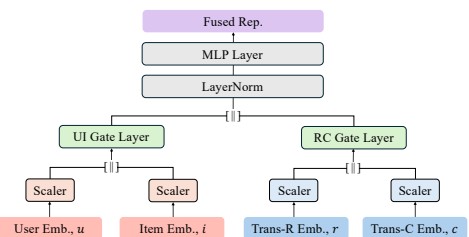

Figure 4: Architecture of the encoder. Here, $[\ \|\ ]$ denotes concatenation along the feature dimension.

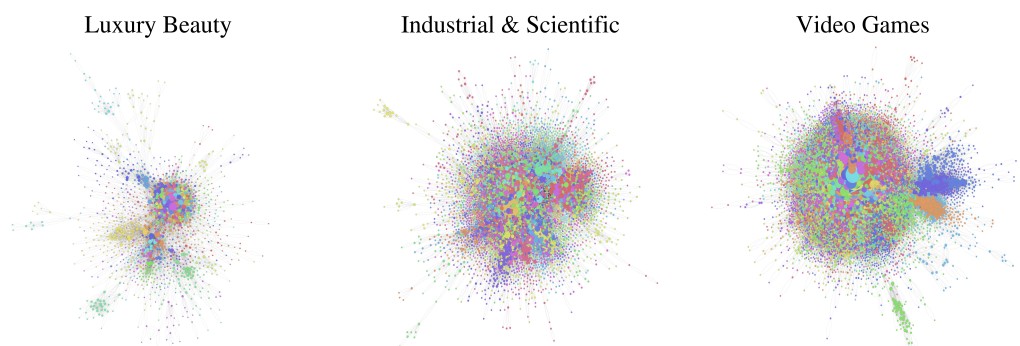

Luxury Beauty · Industrial & Scientific · Video Games

Figure 5: Qualitative visualization of item co-occurrence graphs. All panels share a common orientation and scale; nodes are items (colored by SBM communities), and edges are weighted co-occurrences (thresholded).

gated interaction terms. This design implements a two-stream variant of low-rank bilinear pooling (MLB; Kim et al., 2017), which efficiently approximates full bilinear models and aligns with classical multiplicative interactions in recommender systems (Rendle et al., 2010; Guo et al., 2017). Feature-wise gates provide selective, context-dependent modulation (Perez et al., 2018), enabling adaptive reweighting rather than fixed multiplicative features. The gated outputs from both streams are concatenated, normalized by LayerNorm, and passed through an MLP with GELU activation to yield the final fused representation.

## 5 EXPERIMENTS

**Datasets** We evaluate on three datasets from the Amazon 2018 corpus[1]: Luxury Beauty, Industrial & Scientific, and Video Games. While previous approaches have typically selected datasets solely based on their scale (e.g., the number of users or items), we deliberately choose datasets that differ not only in scale but also in the structural properties of their co-occurrence graphs. Figure 5 visually compares the item co-occurrence graphs across the three datasets. These structural contrasts highlight distinct topological patterns that are likely to reflect varying dataset difficulty and distributional characteristics. Qualitative and quantitative analyses of the co-occurrence graph built by each dataset are summarized in Appendix E. Across all datasets, we convert the data to implicit feedback by treating ratings $\geq 3$ as positive interactions, and we remove users/items with fewer than 5 interactions.

**Baselines** We compare SR-PFN against five representative ID-based and two large language model (LLM)-based sequential recommenders. For ID-based baselines, we include FPMC (Rendle et al., 2010), GRU4Rec (Hidasi et al., 2015), NextItNet (Yuan et al., 2019), Caser (Tang & Wang, 2018), and SASRec (Kang & McAuley, 2018). For LLM-based baselines, we adopt CTRL (Li et al., 2025) and LLM-SRec (Kim et al., 2025). While most LLM-based recommenders focus on text generation, these two emphasize learning representations for recommendation, making them directly comparable to our ranking-based setting. Further details of each baseline are provided in the Appendix F.

**Evaluation protocol** We adopt the widely used *leave-one-out* evaluation protocol (Kang & McAuley, 2018; Sun et al., 2019) for sequential recommendation. For each user sequence, we take all but the last two items for training, the penultimate item for validation, and the final item for testing. Following prior work (Kim et al., 2024; Zhang et al., 2025b), we form the candidate set by including the ground-truth positive item together with 19 randomly sampled negative items that the user has not interacted with. For evaluation, we measure performance using the HR@1 and NDCG@5 metrics to capture both strict top-1 accuracy and position-sensitive ranking quality within the top-5 results. Specifically, HR@1 measures the fraction of cases where the ground-truth next item is ranked first among the candidates, and NDCG@5 evaluates whether it appears within the top-5 while giving higher credit to higher-ranked positions. The maximum sequence length is fixed to 50 for all baselines and our model to ensure a comparable protocol.

Table 1: Results of SR-PFN compared with sequential recommender models. The bold indicates the best performance.

| Dataset | Metric | ID-based | | | | | LLM-based | | PFN-based |
|---|---|---|---|---|---|---|---|---|---|
| | | FPMC | GRU4Rec | NextItNet | Caser | SASRec | CTRL | LLM-SRec | SR-PFN |
| Luxury Beauty | HR@1 | 0.2779 | 0.4005 | 0.4102 | 0.4314 | 0.5035 | 0.2754 | 0.5055 | **0.5222** |
| | NDCG@5 | 0.3658 | 0.5336 | 0.5293 | 0.5543 | 0.5981 | 0.4156 | 0.6412 | **0.6509** |
| Ind. & Sci. | HR@1 | 0.1180 | 0.2132 | 0.2056 | 0.2542 | 0.2695 | 0.1965 | 0.2613 | **0.2894** |
| | NDCG@5 | 0.2069 | 0.3684 | 0.3490 | 0.4014 | 0.4075 | 0.3317 | **0.4395** | 0.4244 |
| Video Games | HR@1 | 0.2565 | 0.4184 | 0.4180 | 0.4551 | 0.5191 | 0.3134 | 0.5238 | **0.5463** |
| | NDCG@5 | 0.4251 | 0.5898 | 0.5796 | 0.6209 | 0.6693 | 0.4931 | **0.6735** | 0.6672 |

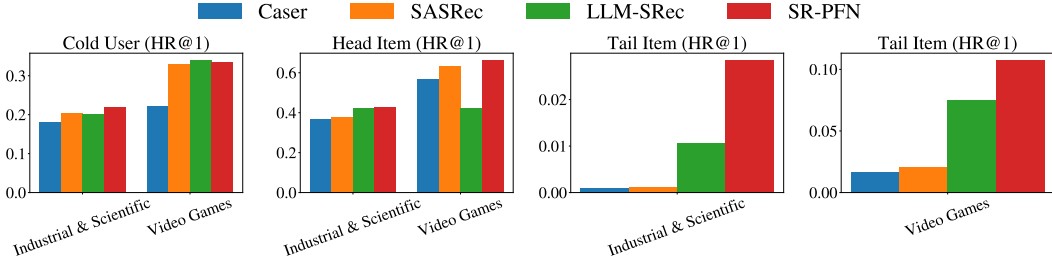

Figure 6: Cold-user and head/tail item performance comparison (HR@1). Results are reported for Industrial & Scientific and Video Games.

## 5.1 MAIN RESULTS

As shown in Table 1, across all three datasets, SR-PFN achieves state-of-the-art performance once pretrained with synthetic priors, consistently surpassing strong ID-based baselines such as SASRec and also outperforming LLM-based methods in terms of top-1 accuracy (HR@1). For instance, on Luxury Beauty, SR-PFN improves over SASRec (0.5222 vs. 0.5035), and on Video Games it surpasses LLM-SRec (0.5463 vs. 0.5238). These gains are notable since SR-PFN is pretrained once in synthetic priors and requires no retraining on the evaluation datasets. Importantly, the main results are reported with in-context examples $k = 4$, which we identified as a sweet spot balancing accuracy and efficiency. In terms of ranking quality, NDCG@5, SR-PFN remains highly competitive, although LLM-SRec occasionally attains slightly higher scores. Overall, these results highlight that once pretrained with synthetic priors, SR-PFN can do inference effectively to unseen datasets only with a small set of examples, offering strong next-item accuracy while maintaining competitive top-$k$ ranking quality, all without the need for billion-scale LLMs.

## 5.2 COLD USER AND HEAD/TAIL ITEM SCENARIOS

**Cold user scenario** We evaluate our approach in the cold-user setting, where models must generalize to users with very limited interaction history. To this end, we construct a test split by selecting users with sequence length exactly three and treat the final interaction as the held-out target item. The results are illustrated in the leftmost panel of Figure 6. On Industrial & Scientific, SR-PFN attains higher HR@1 compared with baselines (Caser, SASRec, and LLM-SRec), while on Video Games, its performance is comparable to the best baseline. These results suggest that the inference mechanism of SR-PFN can leverage structural priors to improve recommendations for cold users, whereas conventional sequential models are less effective in this regime.

**Head/tail item scenario** We further evaluate model behavior with respect to item popularity by reporting performance on head and tail items separately. Items are partitioned based on their empirical popularity in the training data: the top 30% most frequently interacted items are categorized as heads, while the bottom 30% constitute the tails. While SR-PFN is explicitly trained with priors that model the nature of heavy-tailed popularity distributions with head dominance, it exhibits lower

popularity bias than baseline models. As shown in Figure 6, SR-PFN matches or surpasses baselines on head items and yields substantial improvements on tail items, particularly for the Industrial & Scientific dataset. These results suggest that SR-PFN not only adapts to the head-dominated distribution of real-world recommendation data but also preserves accuracy on rare items, exhibiting less popularity bias compared with strong baselines.

## 5.3 INFERENCE SPEED

While LLMs have recently demonstrated strong performance in recommendation tasks, their application typically requires long prompts that incorporate task instructions, user interaction histories, candidate items, and auxiliary context (Wu et al., 2024). This substantially increases inference cost and latency, and the effect is further amplified by model scale, as LLMs used for recommendation generally exceed 3B parameters (Kim et al., 2025). In contrast, SR-PFN is a lightweight 168M-parameter model that provides more computationally efficient inference. To quantify this, we measured inference speed as the number of

Table 2: Inference throughput (higher is better). $k$ denotes the number of in-context examples.

| Model | Throughput (queries s$^{-1}$) |
| --- | --- |
| LLM-SRec | 18.91 |
| SR-PFN ($k = 8$) | 84.97 |
| SR-PFN ($k = 4$) | 126.46 |
| SR-PFN ($k = 2$) | 200.90 |
| SR-PFN ($k = 1$) | 308.03 |
| SR-PFN ($k = 0$) | 491.88 |

queries processed per second with a batch size of 16. Under our basic setting ($k = 4$), SR-PFN achieves 6.69× higher throughput than the LLM-SRec baseline (Table 2), while maintaining comparable performance. In the zero-shot scenario, SR-PFN achieves a throughput of 26× higher than LLM-SRec, even performing better on Industrial & Scientific and Video Games datasets (see Appendix H.2) — making it more efficient for massive user traffic or strict real-time scenarios where speed and efficiency are critical.

## 6 CONCLUSION AND FUTURE WORK

In this work, we introduce SR-PFN, a first attempt to adopt PFNs into the sequential recommendation task. Our key contributions are (i) a controllable synthetic prior that couples an hDCSBM item graph with PPR-based sequence generation, and (ii) an architectural design that extracts sequential patterns from a few example blocks and transfers them to the query. Trained once on diverse synthetic tasks, SR-PFN generalizes across datasets without re-training. It achieves strong accuracy against notable sequential recommender baselines and shows robust performance on cold users, a long-standing challenge for conventional models. Moreover, SR-PFN mitigates popularity bias by producing more balanced recommendations across head and tail items, while also delivering substantially high-throughput inference, making it more suitable for efficient large-scale deployment than LLM-based models.

While our pretraining tasks implicitly emphasize positive interactions, SR-PFN does not explicitly encode negative preferences or avoidance signals. Also, as with other PFN-based approaches, SR-PFN is sensitive to prior misspecification, where its effectiveness depends on how closely the synthetic prior reflects the structure of real-world interactions, and systematic robustness to deviations remains an open question. Finally, although we validated SR-PFN on datasets of up to 50,000 users, scaling to substantially larger catalogs and user bases remains for future work.

## ETHICS STATEMENTS

This work focuses on methodological contributions to the sequential recommendation task. All real-world evaluation datasets used in this study are publicly available subsets of the Amazon product review corpus, which contain no personally identifiable information beyond anonymized user and item identifiers. In terms of computational impact, SR-PFN is substantially more efficient at inference than large language model (LLM)-based recommenders, leading to lower energy consumption per prediction.

## REPRODUCIBILITY STATEMENT

The source code of SR-PFN is available at `https://sites.google.com/view/srpfn-iclr2026/`. Here, we provide an `environment.yml` file to fully specify the software environment. The synthetic data generation process and all associated hyperparameters are documented in detail (see Section 3.2, Appendix C, D). We also report the model architecture, parameter counts, and all training hyperparameters to ensure full reproducibility along with hardware specifications and compute time required for our experiments (see Appendix G).

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

# A   USE OF LLMS

For our work, we used LLMs to polish the writing and to assist coding.

# B   PROOF OF POSTERIOR PREDICTIVE OPTIMALITY FOR SR-PFN

We show that SR-PFN trained with the candidate-restricted cross-entropy converges to the (candidate-restricted) Bayesian posterior predictive distribution (PPD).

## B.1   PROBLEM SETUP

Let $\Phi$ be a hypothesis class with prior $p(\phi)$. Given $\phi$, draw a support set of in-context examples $D = \{(x_i, y_i)\}_{i=1}^n \sim p(D \mid \phi)$ and a query $(x_q, y_q) \sim p(x_q, y_q \mid \phi)$ (independence is not required). Thus $(D, x_q, y_q) \sim \int p(D, x_q, y_q \mid \phi) \, p(\phi) \, d\phi$. For evaluation, each query $x_q$ is scored over a finite candidate set $C_q \subset \mathcal{Y}$ drawn by a negative-sampling policy $\nu(C_q \mid D, x_q, y_q)$ that always includes $y_q$.

**Candidate policy (uniform negatives over the unseen pool)**   Let $U^-(D, x_q)$ be the set of items not previously interacted with by $x_q$ among the dataset $D$. We assume (i) $C_q = \{y_q\} \cup S_q$ with $S_q \subseteq U^-(D, x_q) \setminus \{y_q\}$ and $|C_q| = m$ fixed; (ii) conditional on $(D, x_q)$, $S_q$ is sampled uniformly without replacement from $U^-(D, x_q) \setminus \{y_q\}$. Equivalently, for any fixed $C$ of size $m$ and any $y', y'' \in C$,

$$\nu(C \mid D, x_q, y') = \nu(C \mid D, x_q, y'') = \frac{1}{\binom{|U^-(D, x_q)| - 1}{m - 1}}.$$

## B.2   CANDIDATE-RESTRICTED PPD

Condition on $Z = (D, x_q, C_q)$. By Bayes' rule and the inclusion of $y$ in $C_q$,

$$p(y \mid Z) \propto \mathbf{1}\{y \in C_q\} \, p(y \mid D, x_q) \, \nu(C_q \mid D, x_q, y), \tag{4}$$

which normalizes to

$$p(y \mid D, x_q, C_q) = \frac{p(y \mid D, x_q) \, \nu(C_q \mid D, x_q, y)}{\sum_{c \in C_q} p(c \mid D, x_q) \, \nu(C_q \mid D, x_q, c)}. \tag{5}$$

Under uniform negatives (above), $\nu(C_q \mid D, x_q, y)$ is constant in $y \in C_q$ and cancels, yielding the renormalized form

$$p(y \mid D, x_q, C_q) = \frac{p(y \mid D, x_q)}{\sum_{c \in C_q} p(c \mid D, x_q)} \quad (y \in C_q), \qquad p(\cdot) = 0 \text{ on } \mathcal{Y} \setminus C_q. \tag{6}$$

## B.3   TRAINING OBJECTIVE

SR-PFN is trained by the candidate-restricted cross-entropy

$$\mathcal{L}_{\text{SR}}(\theta) = \mathbb{E}_{(D, x_q, y_q), \, C_q} \big[ -\log q_\theta(y_q \mid D, x_q, C_q) \big], \tag{7}$$

with $(D, x_q, y_q) \sim \int p(\cdot \mid \phi) p(\phi) \, d\phi$ and $C_q \sim \nu(\cdot \mid D, x_q, y_q)$, matching the training interface used at test time.

## B.4   OPTIMALITY WITH INTRA-QUERY CANDIDATE ATTENTION

Allowing full attention among *candidate tokens within the query block* only changes the available conditioning set $Z$; it does not affect the probabilistic identity below. History tokens remain causally masked; ground-truth labels are never input tokens.

**Lemma 1** (CE–KL identity). *For any fixed $Z = (D, x_q, C_q)$, letting $p(\cdot \mid Z)$ denote equation 5,*

$$\mathbb{E}_{y \sim p(\cdot \mid Z)}[-\log q_\theta(y \mid Z)] = H\big(p(\cdot \mid Z)\big) + \text{KL}\big(p(\cdot \mid Z) \,\|\, q_\theta(\cdot \mid Z)\big).$$

*Hence $\mathbb{E}_Z[\cdot]$ of the left-hand side is minimized iff $q_\theta(\cdot \mid Z) = p(\cdot \mid Z)$ almost surely.*

**Proposition 1** (Candidate-restricted optimality). *For each prompt $Z = (D, x_q, C_q)$, the minimizer of equation 7 over distributions supported on $C_q$ is the true conditional $p(\cdot \mid Z)$ in equation 5. Under uniform negatives, this reduces to the renormalized form equation 6.*

**Corollary 1** (Top-1 Bayes decision within $C_q$). *If $y_q \in C_q$ almost surely, then*

$$\arg\max_{c \in C_q} q_\theta^\star(c \mid D, x_q, C_q) = \arg\max_{c \in C_q} p(c \mid D, x_q, C_q) = \arg\max_{c \in C_q} p(c \mid x_q, D).$$

**Remarks** The uniform-without-replacement policy over the unseen pool leaves the relative ranking by the unconditional predictive $p(\cdot \mid D, x_q)$ untouched inside $C_q$. But if the mining policy is popularity-biased or hard-negative (e.g., nearest-neighbor or model-driven mining), the implicit target becomes a reweighted conditional that favors items more likely under the mining policy. This can be desirable for certain head-heavy objectives, but it no longer coincides with the unbiased posterior predictive unless one compensates during training (e.g., by importance weighting or by restoring uniform negatives at training time).

## C  SYNTHETIC DATA GENERATION AND HYPERPARAMETERS

### C.1  HIERARCHICAL COMMUNITY STRUCTURES

Here, we explain how we modeled a hierarchical community structure. We split the item set into a small number of macro-communities, each of which is further divided into several micro-communities. In what follows, the term community refers to a micro-community. We choose the sizes of macros first, and then the sizes of micros within each macro. Both sets of sizes are drawn from power-law distributions so that some groups are significantly larger than others. This allows us to have controllable, imbalanced group sizes. To control how strongly items are linked, we use three block affinities, ordered from strongest to weakest: (i) within the same micro-community, (ii) between different micros of the same macro, and (iii) across different macros. Practically, we sample three nonnegative weights, sort them in descending order, and assign them to these three regimes to enforce the hierarchy by construction. Each item receives its own out-degree and in-degree propensity, drawn from a power-law distribution. The expected connection strength between two items is then determined by (a) their propensities and (b) the affinity between their communities. Using the expected connection strengths, we construct a sparse weighted adjacency matrix and normalize its rows to obtain a Markov transition kernel.

### C.2  STICKINESS

To model stickiness — the tendency of users to remain at the same item — we modify the transition matrix by injecting self-loops. Concretely, we add a self-loop weight to each node in proportion to its degree, controlled by a global coefficient $s$. This makes popular items more likely to exhibit persistence, reflecting the fact that highly connected items are harder to leave. After adding these degree-scaled self-loops, we normalize each row of the matrix so that it defines a valid probability distribution. The resulting transition kernel therefore captures not only the popularity skew and community structure encoded in the original graph, but also a controllable persistence effect that models users staying on the current item.

### C.3  HYPERPARAMETERS

Table 3 summarizes the ranges and distributions of random-sampled hyperparameters used in our synthetic prior construction. These values are re-sampled for each epoch, ensuring diverse graph topologies and sequence statistics. Here, $\text{Uniform}(a, b)$ denotes a continuous uniform draw on $[a, b]$, and $\text{UniformInt}(a, b)$ denotes a discrete uniform draw on the integers $\{a, \ldots, b\}$ (inclusive).

Table 3: Hyperparameters used in the synthetic graph prior and sequence generation.

| Parameter | Sampling dist. | Min | Max |
|---|---|---|---|
| hDCSBM | | | |
| Number of items ($M$) | Uniform() | 3,200 | 25,600 |
| Number of macro blocks ($K_{\text{macro}}$) | UniformInt() | 4 | 12 |
| Number of micro-blocks per macro ($m_j$) | UniformInt() | 2 | 8 |
| Avg. degree ($\bar{d}$) | UniformInt() | 32 | 128 |
| Degree exponent $\gamma_{\text{deg}}$ | Uniform() | 2.0 | 5.0 |
| Within-micro connection weight ($w$)$^\dagger$ | Uniform() | 6.0 | 12.0 |
| Within-macro connection weight ($w$)$^\dagger$ | Uniform() | 1.0 | 4.0 |
| Cross-macro connection weight ($w$)$^\dagger$ | Uniform() | 0.05 | 0.40 |
| Macro-level size exponent ($\tau_{\text{macro}}$) | Uniform() | 1 | 5 |
| Micro-level size exponent ($\tau_{\text{micro}}$) | Uniform() | 1 | 5 |
| *Sequence generation* | | | |
| Number of synthetic users ($U_{\text{synth}}$) | $\max\{16000, \lceil M \cdot \text{Uniform}(1,5) \rceil\}$ | 16,000 | $5M$ |
| Max sequence-length factor ($f_{\text{len}}$) | Uniform() | 0.1 | 1.0 |
| Sequence length power-law exponent ($\alpha_{\text{len}}$) | Uniform() | 1.0 | 2.0 |
| Candidate pool ratio ($r_{\text{pool}}$) | Uniform() | 0.001 | 0.01 |
| PPR restart probability ($\alpha$) | Uniform() | 0.01 | 0.99 |

$\dagger$ Independently sampled, then sorted (descending) and assigned to (within micro, within macro, cross macro).

Table 4: Special tokens used by SR-PFN.

| Token | ID | Role (summary) |
|---|---|---|
| PAD | 0 | Padding token (unused positions) |
| EXAMPLE_START | 1 | Start of an in-context example block |
| CONTEXT_END | 2 | Boundary between history and candidates |
| CANDIDATE_START | 3 | Start of candidate list (attention confined within the block) |
| ANSWER | 4 | Marks the correct candidate *inside examples only* (never in queries) |
| EXAMPLE_END | 5 | End of example block |
| QUERY_START | 6 | Start of query block |
| QUERY_END | 7 | End of query block |
| INTERACTION | -1 | Placeholder for interaction |

## D  IMPLEMENTATION DETAILS

### D.1  SPECIAL TOKENS

The role and summary of each token are given in Table 4. EXAMPLE_START/EXAMPLE_END and QUERY_START/QUERY_END delimit example and query blocks; CONTEXT_END marks the end of the user history and CANDIDATE_START begins the candidate list. ANSWER is inserted immediately before the true positive *only inside examples* and never appears in queries. PAD is used solely for batching and is fully masked out. INTERACTION is a placeholder that serializes to actual item IDs at write time (i.e., it is not a fixed special token). During training and evaluation, scores and loss are computed *only* over the query's candidates. To prevent information leakage, the true query item $y_q$ never appears as ANSWER, and items that will later serve as query ground truth are excluded when constructing interaction/transition matrices and graphs.

### D.2  IN-CONTEXT EXAMPLE SELECTION ALGORITHM

For a query user $q$, we form a candidate user pool $\mathcal{U}_q$ that might be potentially selected as in-context examples. We represent $q$ and each candidate user $i \in \mathcal{U}_q$ by $\ell_2$-normalized embeddings $\mathbf{u}_q$ and $\mathbf{u}_i$, and define their *relevance score* as $r_i = \langle \mathbf{u}_q, \mathbf{u}_i \rangle$. By selecting the most relevant candidate user as

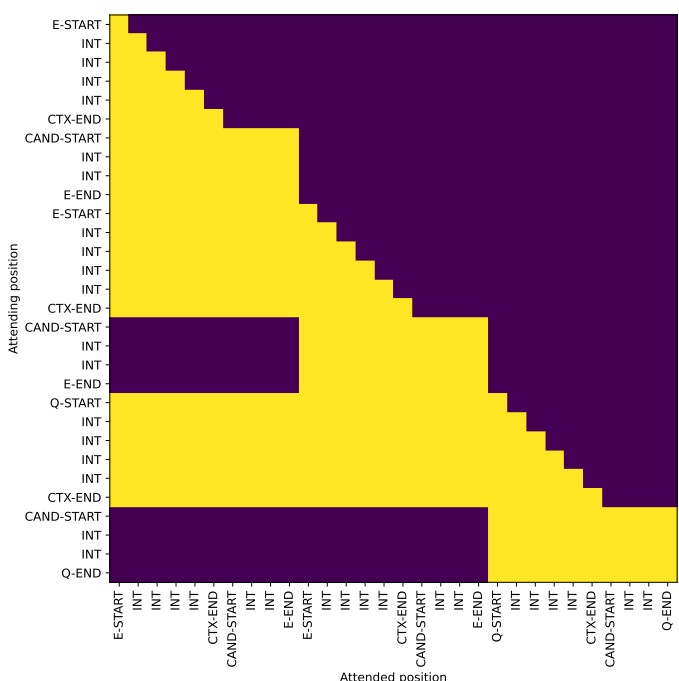

Figure 7: Visualization of the attention mask with two examples and one query.

the first example, we iteratively select at step $t$,

$$i^\star \;=\; \arg\max_{i \in \mathcal{U}_q \setminus S_{t-1}} \left\{ \lambda\, r_i \;-\; (1-\lambda)\, d_i \right\},$$

where $d_i \;=\; \max_{j \in S_{t-1}} \langle \mathbf{u}_i, \mathbf{u}_j \rangle$ encodes how redundant candidate $i$ is with the already–chosen exemplars. After adding $i^\star$ to the set $(S_t \leftarrow S_{t-1} \cup \{i^\star\})$, we update the penalties for the remaining candidates as

$$d_i \;\leftarrow\; \max\big\{ d_i,\; \langle \mathbf{u}_i, \mathbf{u}_{i^\star} \rangle \big\} \quad \text{for all } i \in \mathcal{U}_q \setminus S_t.$$

In this way, $r_i$ measures how well candidate $i$ matches the query user $q$, while $d_i$ reflects how redundant it is with the current exemplar set. The trade-off parameter $\lambda \in [0, 1]$ balances the two: $\lambda = 1$ selects purely by relevance (top-$k$ by $r_i$), while $\lambda = 0$ enforces maximal diversity (farthest-first sampling). Iterating this procedure until $k$ exemplars are chosen yields a set that is both relevant to the query and diverse among themselves, thereby providing richer in-context information without unnecessary repetition. In this work, the hyperparameter $\lambda$ is fixed to $0.5$.

### D.3 VISUALIZATION OF ATTENTION MASK

Figure 7 shows a attention mask represented as binary matrix $M \in \{0,1\}^{T \times T}$ for a prompt with two *example* blocks and one *query* block, each having a history of length 5 and a candidate list of size 3 (allowed attention = yellow; disallowed = dark).

We construct $M$ as the conjunction of an inter-block and an intra-block policy: (i) *Inter-block:* tokens inside example blocks may attend only within their own and earlier example blocks (no look-ahead across examples), while tokens inside the query block may attend to all preceding blocks as well as within the query block. (ii) *Intra-block:* each block is split at CONTEXT_END into a history region and a candidate region. History tokens use a left-to-right causal mask, hence the triangular pattern inside each history segment. Candidate tokens have full attention *within the same block* (rectangular patches), so they can attend to the block's history and to other candidates, but never across blocks.

Table 5: Dataset statistics after preprocessing (5-core).

| Dataset | #Users | #Items | Avg. Seq. Len. |
|---|---|---|---|
| Luxury Beauty | 5,198 | 5,120 | 8.54 |
| Ind. & Sci. | 24,831 | 26,109 | 6.56 |
| Video Games | 52,944 | 30,355 | 9.50 |

Table 6: Topology of item co-occurrence graphs.

| Dataset | $N$ (items) | $E$ (edges) | Density | $\bar{d}$ | Transitivity | Hub dom. | Non-GC frac. |
|---|---|---|---|---|---|---|---|
| Luxury Beauty | 5,120 | 23,416 | $1.79 \times 10^{-3}$ | 9.15 | 0.266 | 11.6% | 0.666 |
| Ind. & Sci. | 26,109 | 19,710 | $5.78 \times 10^{-5}$ | 1.51 | 0.110 | 2.3% | 0.804 |
| Video Games | 30,355 | 452,353 | $9.82 \times 10^{-4}$ | 29.80 | 0.208 | 6.8% | 0.430 |

Formally, $M_{ab} = M_{ab}^{\text{inter}} \wedge M_{ab}^{\text{intra}}$. We additionally maintain a binary indicator $m^{\text{qc}} \in \{0, 1\}^T$ that restricts scoring and loss computation *only* to the candidate span in the query block; example candidates are never scored.

# E  DATASET STATISTICS AND CO-OCCURRENCE ANALYSIS

## E.1  SEQUENCE-LEVEL STATISTICS

Table 5 reports the number of users and items and the average sequence length (including the held-out validation/test item) after 5-core filtering. Concretely, Video Games has the longest sequences on average (9.50) and the largest user base (52,944), whereas Industrial & Scientific offers the shortest sequences (6.56) with a mid-sized catalog (26,109 items); Luxury Beauty sits in between in terms of length (8.54) but with the smallest catalog (5,120 items).

## E.2  QUALITATIVE INTERPRETATION OF CO-OCCURRENCE GRAPHS

Figure 5 visually compares the item co-occurrence graphs across the three datasets. Luxury Beauty exhibits a clear hub-and-spoke structure centered on a dominant core, accompanied by several medium-sized communities connected through bridges. Industrial & Scientific appears more diffuse, characterized by a compact giant component but weaker modularity. In contrast, Video Games presents the densest core with pronounced hub dominance and elongated filamentary connections. These structural contrasts anticipate the quantitative differences reported in Table 6, highlighting distinct topological patterns that reflect varying dataset difficulty and distributional characteristics.

## E.3  QUANTITATIVE INTERPRETATION OF CO-OCCURRENCE GRAPHS

**Metrics**  On the co-occurrence graph built by each dataset, we report:

- **Density** $= \dfrac{2E}{N(N-1)}$ — share of realized item–item links ($\in [0, 1]$).

- **Average degree** $\bar{d} = \dfrac{2E}{N}$ — typical # of co-occurring neighbors per item.

- **Transitivity** (global clustering) — fraction of closed triads; higher means stronger local closure.

- **Hub dominance** $= \dfrac{k_{\text{max}}}{N-1}$ — largest hub's reach as a share of items.

- **Fragmentation** $= 1 - f_{\text{GC}}$ — fraction of items outside the giant component.

As shown in Table 6, Industrial & Scientific is very sparse and fragmented, having low density and average degree with high non–GC fraction. Meanwhile, Luxury Beauty is more connected with stronger local closure (higher transitivity) and a more pronounced single hub. Video Games has a

Table 7: Model configuration of SR-PFN.

| Parameter | Value |
|---|---|
| Embedding dimension ($d$) | 1024 |
| Number of layers | 12 |
| Hidden dimension ($4d$) | 4096 |
| Number of attention heads | 16 |
| Dropout rate | 0.1 |
| Activation function | GELU |
| Input normalization | True |

large, dense giant component (the highest $\bar{d}$ here) with moderate hub concentration and the least fragmentation among the three. Together, these results show that the datasets differ not only in sequence volume but also in connectivity patterns, motivating models that remain reliable across diverse graph regimes.

## F  BASELINES

We compare SR-PFN against five representative ID-based and two language model(LM)-based sequential recommenders that leverage semantic information beyond ID representations. FPMC (Rendle et al., 2010) factorizes user-item preferences while coupling them with a first-order Markov chain to capture short-term transitions. GRU4Rec (Hidasi et al., 2015) models sequences with gated recurrent units and ranking-oriented losses for session-based recommendation. NextItNet (Yuan et al., 2019) replaces recurrence with deep stacks of dilated causal convolutions to encode long-range dependencies. Caser (Tang & Wang, 2018) uses horizontal and vertical convolutional filters to extract union-level and point-level sequential patterns. SASRec (Kang & McAuley, 2018) applies unidirectional self-attention to learn variable-order item dependencies. CTRL (Li et al., 2025) reformulates the recommendation task as a text-prompt and aligns semantic representations with a CF model. Most recent LLM-based recommenders are formulated as generative tasks that output a single target item, making them unsuitable for direct comparison with our ranking-based setting. We therefore adopt LLM-SRec (Kim et al., 2025) as our LLM baseline, which distills representations from a sequential CF model to better capture user preferences.

## G  TRAINING SETUP AND HYPERPARAMETERS

### G.1  TRAINING SR-PFN

The model configuration of SR-PFN is summarized in Table 7. In total, the model contains approximately 168M trainable parameters when the the SVD dimension is set to 1024. We trained SR-PFN for a total of 500 epochs, each epoch comprising up to 1,000 steps (500,000 steps in total) with a batch size of 16. For each epoch, a new synthetic graph was generated; if the graph did not contain sufficient sequences to fill 1,000 steps, additional graphs were sampled to complete the epoch. For Table 1, we set the learning rate to $3 \times 10^{-5}$, the number of in-context examples $k$ to 4, and the SVD dimension to 1024. The total training required approximately 60 GPU hours on a single NVIDIA RTX A6000.

### G.2  OPTIMIZATION DETAILS

To stabilize optimization, we use the AdamW optimizer with gradient clipping, and control the learning rate through a cosine scheduler with warmup. The global gradient norm is clipped to 10 at each update step. Given 500 epochs and 1,000 steps per epoch with gradient accumulation of 16, the total number of optimizer updates is approximately 31,250. The learning rate is linearly warmed up during the first 25 epochs (1,562 updates), and subsequently decays following a cosine schedule down to $10\%$ of the peak value. This schedule enables stable convergence while mitigating sharp drops in training loss. The complete set of training hyperparameters is summarized in Table 8.

Table 8: Training hyperparameters for SR-PFN.

| Hyperparameter | Value |
|---|---|
| Epochs | 500 |
| Steps per epoch | 1,000 |
| Batch size | 16 |
| Gradient accumulation | 16 |
| Learning rate | $\{3 \times 10^{-5}, 5 \times 10^{-5}, 1 \times 10^{-4}\}$ |
| Warmup epochs | 25 |
| Weight decay | $1 \times 10^{-4}$ |
| Optimizer | AdamW |
| Scheduler | Cosine annealing with linear warmup |
| Minimum LR ratio | 0.1 (relative to peak LR) |
| Gradient clipping | 10.0 (global norm) |
| Mixed precision (AMP) | Enabled |

Table 9: Ablation on the embedding dimension $d$ (HR@1).

| | $d{=}256$ | $d{=}512$ | $d{=}1024$ |
|---|---|---|---|
| Luxury Beauty | 0.4900 | 0.5077 | **0.5222** |
| Industrial & Scientific | 0.2777 | 0.2867 | **0.2894** |
| Video Games | 0.5445 | 0.5417 | **0.5463** |

# H  ABLATION STUDIES

## H.1  ABLATION ON EMBEDDING DIMENSION

We vary $d \in \{256, 512, 1024\}$ under an otherwise identical training setup (only the SVD rank / embedding size changes). Larger $d$ yields small but consistent gains on Luxury Beauty and Industrial & Scientific, and Video Games also attains its best performance at $d{=}1024$ with only a marginal improvement over smaller dimensions. We adopt $d{=}1024$ for the main results: it is best or near-best across all datasets and offers a reasonable trade-off between accuracy and capacity.

## H.2  ABLATION ON NUMBER OF IN-CONTEXT EXAMPLES

Table H.2 reports the ablation results with varying numbers of in-context examples $k$. Interestingly, the model already demonstrates non-trivial performance even when $k = 0$, showing that the pre-trained prior itself captures substantial sequential patterns without the aid of examples. The sweet spot is observed at $k = 4$, where the model achieves the best performance in two out of three datasets, suggesting that a moderate number of examples provides sufficient guidance without overwhelming the model. In contrast, using too many examples ($k = 8$) slightly degrades performance, implying that excessive context introduces noise or redundancy that hinders effective generalization.

Table 10: Ablation on the number of in-context examples $k$ (HR@1).

| Dataset | $k = 0$ | $k = 1$ | $k = 2$ | $k = 4$ | $k = 8$ |
|---|---|---|---|---|---|
| Luxury Beauty | 0.5044 | 0.5144 | 0.5123 | **0.5222** | 0.5144 |
| Ind. & Sci. | 0.2917 | **0.2952** | 0.2930 | 0.2894 | 0.2885 |
| Video Games | 0.5445 | 0.5332 | 0.5376 | **0.5463** | 0.5452 |

