# OpenReview forum: "SR-PFN: Yet Another Sequential Recommendation Paradigm"
_ICLR.cc/2026/Conference — ICLR 2026 Conference Withdrawn Submission_

### Official Review · Reviewer_7SoQ · 2025-10-31

**Soundness:** 2
**Presentation:** 2
**Contribution:** 2
**Rating:** 2
**Confidence:** 4

**Summary:**

This paper presents SR-PFN, a sequential recommendation model that operates under the Prior-data Fitted Networks (PFNs) paradigm. The model is pre-trained on synthetic data generated from a prior designed to reflect properties of real-world interaction sequences.  Empirical results on benchmark datasets indicate that this approach can achieve competitive performance compared to several ID-based and LLM-based baselines while offering lower inference costs.

**Strengths:**

- The paper introduces SR-PFN, a novel sequential recommendation model based on the Prior-data Fitted Networks (PFNs) paradigm.

- It features a controllable synthetic prior that captures real-world data properties.

- The figures in this paper are relatively exquisite and clear.

**Weaknesses:**

- The motivation of this paper is insufficient, as it fails to explain the rationale for using PFN in sequential recommendation. It seems to employ this method merely for the sake of using a different approach, which I find unreasonable.

- The introduction mentions that existing LLM methods often incur prohibitive inference costs and latency, limiting their practicality in real-world deployment scenarios. However, many current LLM-based methods do not involve LLM computation during the inference stage, meaning LLM-related calculations are only utilized during training. Therefore, the authors' claim is not objective.

- The experimental results indicate that the performance improvement of the proposed method is quite limited. As shown in Table 1, the method fails to achieve SOTA in NDCG@5 on two datasets. Moreover, the evaluation metrics used in the experiments are inconsistent with those in mainstream literature, as evidenced by the absence of standard indicators such as HR@5, HR@10, and NDCG@10.

- The authors state in the introduction that their method can "make contextually relevant predictions on new tasks by processing examples from real-world datasets in-context, without requiring data-specific training." However, the experimental section lacks relevant validation for this claim. Only cold-user experiments are provided, which are insufficient to substantiate this assertion.

- All three experimental datasets are sourced from Amazon datasets, lacking diversity and variation in sparsity levels.

- In the efficiency comparison presented in Table 2, the selected LLM-based methods do not include those that avoid LLM computation during inference. There are numerous such methods that only incorporate LLM-derived semantic information during training, making their inference speed comparable to ID-based backbones. Furthermore, the authors fail to include the inference speed of ID-based methods in the comparison, making this evaluation unfair and incomplete.

**Questions:**

Please refer to Weaknesses.

---

### Official Review · Reviewer_wjFg · 2025-10-31

**Soundness:** 3
**Presentation:** 3
**Contribution:** 3
**Rating:** 6
**Confidence:** 5

**Summary:**

The authors propose a novel method for sequential recommendation systems, referred to as SR-PFN, which introduces a synthetic prior model tailored for sequential recommendation tasks. This solution marks the first attempt to address sequential recommendation within the Prior-data Fitted Networks (PFNs) setting. The model presented in the paper is logically sound, as it seeks to approximate the posterior predictive distribution (PPD) for next-item prediction following pre-training on synthetic data that is sampled from the prior model. Furthermore, the paper includes comprehensive experiments: it conducts extensive tests across multiple public datasets and one industrial dataset to validate the method’s effectiveness. Additionally, the use of advanced baselines provides solid support for evaluating the experimental results.

**Strengths:**

1. The paper provides a clear and insightful explanation of the proposed method, SR-PFN, and its associated benefits. The motivation behind the paper is well reflected in both the methods and experiments.

2. This method marks the first attempt to address sequential recommendation within the framework of Prior-data Fitted Networks (PFNs).

3. The method disentangles mobility patterns and user preferences, introducing contrastive learning to model user preferences for the first time.

**Weaknesses:**

1. The experimental evaluation only adopts two metrics, HR@1 and NDCG@5, which are somewhat insufficient. It would be beneficial to incorporate additional metrics to provide a more comprehensive assessment of the model’s performance.

2. In the inference speed experiment, the baseline is limited to one LLM-based model, making the comparison somewhat inadequate. Expanding the set of baselines with more representative models would help better validate the proposed method’s speed advantage.

**Questions:**

All raised questions and suggestions have been pointed out in the "Weaknesses" section of our paper. These questions are for reference only:

1. The evaluation only reports HR@1 and NDCG@5 — would it be possible to include more metrics for a more comprehensive performance assessment?

2. In the inference speed analysis, only one LLM-based baseline is used — can more representative models be included to make the comparison more convincing?

---

### Official Review · Reviewer_9SZZ · 2025-11-01

**Soundness:** 3
**Presentation:** 3
**Contribution:** 3
**Rating:** 4
**Confidence:** 4

**Summary:**

The paper, SR-PFN, introduces the novel concept of applying Prior-Data Fitted Networks (PFNs) to sequential recommendation, aiming to offer an alternative that is highly efficient (up to 6.69× higher throughput than the LLM-SRec baseline) and achieves state-of-the-art HR@1 accuracy on several benchmarks. Despite its methodological novelty and strong efficiency metrics, the paper suffers from critical methodological uncertainties and incomplete empirical validation that preclude acceptance. The core weakness is the PFN paradigm's acknowledged sensitivity to prior misspecification, as the model's effectiveness depends critically on how closely the synthetic prior (based on hDCSBM and PPR) reflects real-world structures, leaving the systematic robustness to deviations as an open question. This fundamental uncertainty regarding the model's validity constitutes a major methodological flaw. Furthermore, while the model is positioned for efficient large-scale deployment, the authors admit that scaling to substantially larger catalogs and user bases remains for future work, having only validated the approach on datasets up to approximately 50,000 users. This major omission represents a lack of evidence to support the crucial claims regarding real-world scalability and efficacy.

**Strengths:**

1. Pioneering Methodological Concept: The work represents the first attempt to extend the Prior-Data Fitted Network (PFN) paradigm to sequential recommendation. This framework offers a potentially valuable path to generalization, as the model is pretrained once on synthetic data and performs parameter update-free, single-pass inference.

2. Exceptional Inference Efficiency: SR-PFN is a lightweight model (approximately 168M parameters) that provides substantially lower inference costs compared to LLM-based recommenders. It achieves 6.69× higher throughput than the LLM-SRec baseline in the standard setting (k=4), making it appealing for resource-constrained environments.

3. Sophisticated Prior Model Design: The authors utilized a specialized generative mechanism, coupling a hierarchical degree-corrected stochastic block model (hDCSBM) for item graph generation with Personalized PageRank (PPR) for sequence generation. This approach successfully matches key real-world statistical properties, such as heavy-tailed popularity and hierarchical structure.

**Weaknesses:**

1. Lack of Evidence to Support Scalability Claims (Major Flaw): While the paper champions SR-PFN's suitability for "efficient large-scale deployment", the authors explicitly state that they only validated the model on datasets up to 50,000 users, and scaling to substantially larger catalogs and user bases remains for future work. This represents a crucial lack of evidence to support conclusions regarding the practical applicability of the model at scale, which is essential given its positioning against LLMs.

2. Unaddressed Methodological Fragility (Sensitivity to Prior Misspecification): The paper acknowledges that SR-PFN, like other PFN-based approaches, is sensitive to prior misspecification. Since the model relies entirely on the synthetic prior reflecting real-world interactions, and the systematic robustness to deviations remains an open question, the methodology is inherently fragile. Without demonstrating robustness across a systematic evaluation of prior deviations, the study design is questionable, as it relies on an unverified foundational assumption.

3. Sub-optimal Ranking Performance: Despite achieving high HR@1, the model's overall ranking quality, measured by NDCG@5, is only described as "highly competitive," with LLM-SRec occasionally attaining slightly higher scores. If the primary competitive advantage over LLMs is inference speed, the model must demonstrate undeniable ranking superiority to justify its new paradigm over fine-tuned LLMs when ranking quality matters.

4. Incomplete Preference Modeling: The current model design focuses implicitly on positive interactions through its sequence generation prior. SR-PFN does not explicitly encode negative preferences or avoidance signals. This omission suggests an incomplete model of user preferences, limiting its descriptive power compared to real-world systems that rely on implicit negative feedback.

5. Instability with Context Length: The ablation study revealed a stability issue: while the optimal context was found at k=4, using an excessive context (k=8) slightly degrades performance. This indicates a potential fragility in the in-context learning mechanism, suggesting that the model struggles to effectively filter noise or redundancy when given more examples than necessary.

**Questions:**

NA

---

### Note · Authors · 2025-12-03

**Comment:**

We thank the reviewers and the area chair for their time and constructive feedback. After internal discussion, we have decided to withdraw the submission.

**Withdrawal Confirmation:**

I have read and agree with the venue's withdrawal policy on behalf of myself and my co-authors.